# Sample size requirements for genetic studies on yellowfin tuna

**Scott D. Foster** [1]*, **Pierre Feutry**[2], **Peter Grewe**[2], **Campbell Davies**[2]

**1** CSIRO's Data61, Hobart, Tasmania, Australia, **2** CSIRO's Oceans and Atmospheres, Hobart, Tasmania, Australia

* scott.foster@data61.csiro.au

**Data Availability Statement:** All relevant data are within the manuscript and its Supporting information files.

**Funding:** The authors received no specific funding for this work.

## Abstract

In population genetics, the amount of information for an analytical task is governed by the number of individuals sampled and the amount of genetic information measured on each of those individuals. In this work, we assessed the numbers of individual yellowfin tuna (*Thunnus albacares*) and genetic markers required for ocean-basin scale inferences. We assessed this for three distinct data analysis tasks that are often employed: testing for differences between genetic profiles; stock delineation, and; assignment of individuals to stocks. For all analytical tasks, we used real (not simulated) data from four sampling locations that span the tropical Pacific Ocean. Whilst spatially separated, the genetic differences between the sampling sites were not substantial, a maximum of approximately $F_{st} = 0.02$, which is quite typical of large pelagic fish. We repeatedly sub-sampled the data, mimicking a new survey, and performed the analyses. False positive rates were also assessed by re-sampling and randomly assigning fish to groups. Varying the sample sizes indicated that some analytical tasks, namely profile testing, required relatively few individuals per sampling location ($n \gtrsim 10$) and single nucleotide polymorphisms (SNPs, $m \gtrsim 256$). Stock delineation required more individuals per sampling location ($n \gtrsim 25$). Assignment of fish to sampling locations required substantially more individuals, more in fact than we had available ($n > 50$), although this sample size could be reduced to $n \gtrsim 30$ when individual fish were assumed to belong to one of the groups sampled. With these results, designers of molecular ecological surveys for yellowfin tuna, and users of information from them, can assess whether the information content is adequate for the required inferential task.

## 1 Introduction

Making sure that enough data is gathered to effectively answer specific scientific questions has been recognised to be an important part of the scientific process (e.g. power analysis, see [1]). Without adequate data quantities, results of statistical analyses are likely to be uncertain, in that the result may be prone to high rates of incorrect inferences (e.g. a type I or type II error in hypothesis testing; [1, 2]). The issue of adequate data is of sufficient importance that such considerations are mandatory when proposing surveys / experiments that utilise animal subjects [3], and this importance is paralleled in population genetics.

**Competing interests:** The authors have declared that no competing interests exist.

There have been a number of studies that investigated sample size considerations in molecular ecology, which all had a different taxonomic focus: from plants [4–6], insects [7], birds [7], mammals [7, 8], and synthetic theoretical populations [6, 9–11]. These studies also had a different analytical foci and either try to: 1) estimate allele frequencies [7]; 2) differentiate populations [5, 8, 11, 12]; 3) detect relationships in populations [4, 6, 9], or; 4) infer properties within a population [10]. In genetic studies, the number of individuals is not the only consideration as the number of markers will also affect the information content in the data [12, 13]. This provides another delineating factor for sample size calculations: whether the number of those markers is fixed [4, 7, 8] or is part of the sample size investigation [5, 6, 12]. Investigating the number of markers allows us to quantify the benefit from using modern marker technologies that cheaply provide many (1000s) markers. Such multi-faceted delineation of genetic data, organism and inferential goals makes it difficult to supply generalised sample size advice.

For yellowfin tuna (*Thunnas albacares*), guidelines for data quantities for genetic population studies are lacking. This is in spite of its enormous economic importance and value as a food source. Indeed, we are unaware of published studies aimed at assessing sample size and genotyping requirements for any large pelagic fish, from which we can extrapolate to yellowfin tuna. The inadequacy has meant that sample sizes can only be guessed at, based on information from model organisms which may have experienced quite different evolutionary processes, different life stages, and even different genomic structure. This problem is amplified when different analytical tasks are additionally considered. For tunas, interest is often in testing for population differentiation using genetic profiles (a hypothesis test, [14]), in delineating stocks [15–17], and in assigning individual fish to sampling locations [18, 19]. We refer to these three separate analytical tasks as: genetic profile testing, stock identification, and individual assignment.

In this work, we studied the sample size requirements for future genetic studies on yellowfin tuna. Our approach was to exploit data holdings from sample locations across the species' distribution in the Pacific Ocean, and we repeatedly use these as a resource from which new re-sampled survey data were generated. This approach, while not available for organisms with limited data, should have guaranteed that the re-sampled survey data matched data that is likely from real observations. The same guarantee could not be made for simulated genetic data as information was lacking concerning life histories, evolutionary events, linkage maps, and so on. In this analysis, the re-sampled data sets were used as inputs to analyses for profile testing, stock identification and individual assignment. To understand the data requirements, we varied the size of the re-sampled data in terms of the number of fish and the number of genetic markers measured on each of those fish.

## 2 Methods

To perform this study, three things were needed: 1) quality yellowfin tuna genetic data that was bigger than that required for statistical analyses and also exhibited genetic variation similar to experimental goals, 2) a resampling scheme that produced new *in silico* samples from the original data, and 3) an implementation of the statistical methods that directly matched the desired inferential purposes. The basic idea of our approach was to generate numerous re-samples [20, 21], analyse these re-samples and store the results. In this way, we generated an estimate of the effectiveness of the analyses using sensible assessments of their performance (e.g. p-values for hypothesis tests). This procedure is outlined in Fig 1. Details of the statistical methods used to assess the information content are given in Section 2.3. In the current study, we varied both the numbers of fish sampled and the number of genetic markers scored on each fish. Doing so enabled assessment of whether the analyses had better performance with

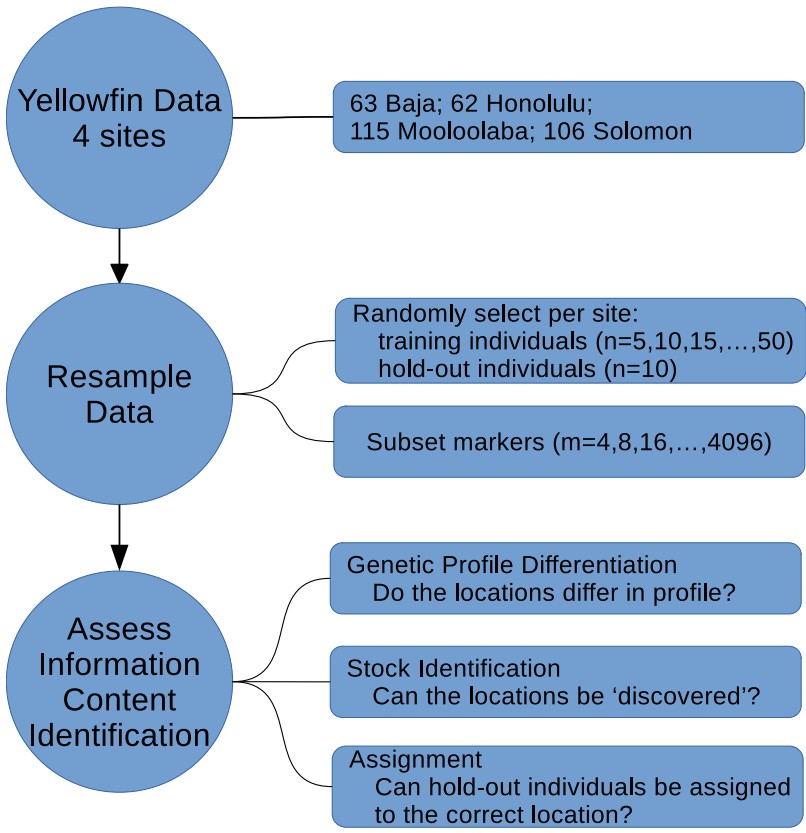

**Fig 1. Depiction of the resampling process used in this study.** Details of the data, resampling strategy, and the statistical methods are given in Section 2.

either sample size delineator, or with both. Our subsampling process mirrored that in [6] except that we considered more combinations of the numbers of individuals and the numbers of markers. This enabled a more detailed investigation of the way that the analytical results responded to combinations of both individuals and markers. Also, we used hold-out samples to assess some inferences (e.g. stock identification).

## 2.1 Yellowfin tuna data

We utilised data on yellowfin tuna (*Thunnus albacares*) from four locations throughout the tropical Pacific Ocean. The locations were (from west to east): Mooloolaba (Australia), Solomon Islands, Hawaii (U.S.A.), and Baja California (Mexico), See Fig 2. These four sites were expected to exhibit small to moderate genetic differences (see Table 1), with the Baja California location being distinct [17, 22].

Each fish was genotyped using single nucleotide polymorphisms (SNP) markers obtained with DArTseq, a Restriction site-associated DNA (RAD) genotyping by sequencing approach [23]. We considered it likely that the SNP data were error prone and noisy, like all genetic data [24]. We tried to mitigate the effect of these errors on inferences and we 'cleaned' or 'filtered'

# Yellowfin Tuna Sampling Areas

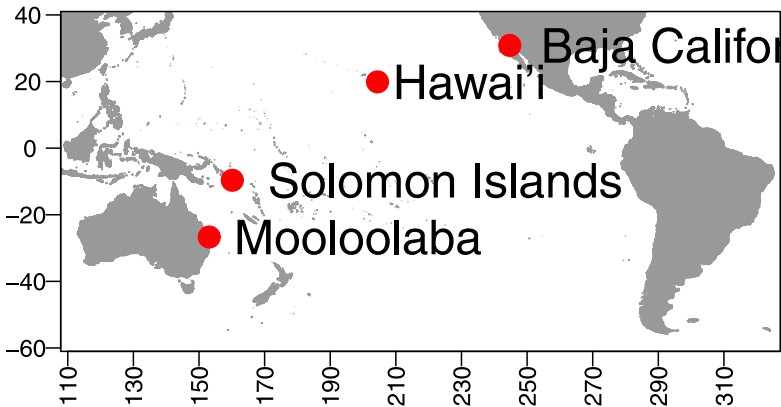

**Fig 2. Approximate sample locations where yellowfin tuna Thunnus albacares were collected.** The three most western areas (Mooloolaba, Solomon Islands and Hawai'i) showed small genetic differentiation and were sometimes separated as the "Western Pacific" data.

the markers data to remove obviously erroneous values. This was performed using the radiator R-package [25], which removes low quality markers/individuals without biasing the molecular signal in the data. We removed markers exhibiting low reproducibility, low minor allele frequencies (MAF), low/high read depths, high missingness, and frequencies not in Hardy-Weinberg equilibrium. For each locus only the marker with the highest MAF was retained. We also removed fish that were highly/poorly heterozygous, or had genotypes that were very similar to other fish (duplicated). The dataset was checked for monomorphic markers after each filtering step that removed individuals. In addition, we removed SNP markers with minor alleles observed in less than 5 fish or that were observed fewer than 2 times within each sampling location. After filtering we retained 6051 SNP markers on 346 yellowfin tuna.

To avoid complications with some, but not all, analytical methods we imputed missing values and created a 'complete' data set. We assumed that, within a sampling location, the missing values were missing completely at random [26, Chapter 25] and performed a simple imputation by randomly drawing a fish's missing marker value based on the allele frequency from an individual's sampling location. The end result of this process was a complete data set, whose entries were either observed or were a random realisation of the observed marker data. In total there were ∼2.3% values imputed, out of a total of 2,044,830 values.

## 2.2 Resampling approach

The resampling process is outlined in Fig 1. The observed data from the four sites were resampled so that there were the same numbers of fish per population sample. A 'hold-out' sample

**Table 1. Sample sizes (top row) and pairwise Fst for the different geographical locations.** Fst values were calculated after cleaning of the genetic data and imputation of missing values (see Section 2.1).

| | Mooloolaba | Solomon Islands | Hawaii | Baja California |
|---|---|---|---|---|
| Sample size | 115 | 106 | 62 | 63 |
| Mooloolaba | — | | | |
| Solomon Islands | 0.0005 | — | | |
| Hawaii | 0.0022 | 0.0024 | — | |
| Baja California | 0.0205 | 0.0202 | 0.0129 | — |

was also generated for assessing the predictive performance of the statistical methods, where it was appropriate to do so. As an example, a resample of 20 Mooloolaba fish was randomly sampled (without replacement) of the 115 real fish, and the 10 hold-out fish were randomly chosen from the remaining 95 fish.

In addition to sampling fish, a subset of markers was also taken randomly from the full set of markers. The reason for using a random sample, and not a sample of the most informative markers, was, when scoring markers, we usually do not know *a priori* if it was going to be informative or not.

To understand the effects of increasing/decreasing sample sizes, and the number of markers, on the amount of information in the data, we varied the number of fish and markers resampled. We ranged the number of fish sampled from $n = 5$ per population to $n = 50$ per population, in increments of 5. We stopped at $n = 50$ so that there remained at least $n = 10$ per sample location for a hold-out sample to assess the out-of-sample assignment performance. Similarly, we incremented the number of markers according to a doubling sequence; $m = 4,8,16,\ldots,4096$. At each combination of sample size and number of markers, 25 data sets were resampled and analysed using the methods below. The results were a summary of those 25 resampled data sets.

To understand if the results were spurious or real, we analysed data sets which *have no genetic structure*. This was done by taking the West Pacific data only (excluding Baja California) and randomly assigning fish to one of the 3 sampling sites (see Table 1). Some researchers may be familiar with this idea as, for statistical hypothesis testing, analysing these randomised data sets gives an estimate of rates of false-positive (type I) errors. For other types of analyses the idea of false positives is not applicable, but the intuition still applies: how strong an inference can be wrongly made.

## 2.3 Statistical analyses

Three different analyses were performed on each resampled data set. The analyses were chosen to reflect questions central to management of wild populations. Details will be presented below, but these analyses can be described as:

**Genetic profile delineation** Was there a difference between the genetic profiles (allele frequencies) of the different locations? This hypothesis test gave a p-value.

**Stock Identification** Could the sampling locations (of individual fish) be reconstructed as the major sources of variation? This was a stock-structure analysis.

**Individual Assignment** Could an individual fish be assigned to its sampling location? Was there enough information to assign them to *any* location?

**2.3.1 Genetic profile delineation.** This analysis tested if the allele frequencies amongst two or more previously defined groups were different in relation to noise in the data (locations in these data). We called this a hypothesis test as we were formally comparing the likelihood of a null hypothesis in relation to an alternative hypothesis. Formally, the null hypothesis was that there is no difference between the groups, and the alternative hypothesis was that there is a difference. We note that this difference could be that just one location differed from the others, or that they all differed from each other. The statistical method used on the genetic marker data are those described in [14], which is well-known as Analysis of MOlecular VAriance (AMOVA).

For each resampled data set the AMOVA analysis produced a p-value. To present the results, these p-values were averaged and then plotted as a colour map with axes of: 1) the number of fish sampled, and 2) the number of markers used. To provide further information on analysis performance, we tested an extra two hypotheses. The first was to see if the Baja California population could be distinguished from the remaining three populations (whose genetic profiles are assumed to be equal). This test was performed as the Baja California population has been shown previously to be quite different [22] and should therefore require fewer samples. The second test was to see if there were any differences between the Honolulu, Solomon Islands, and Mooloolaba populations. This test was performed to find out how much information is needed to separate populations with relatively small genetic differences.

We repeated the hypothesis test on the Western Pacific data (those excluding Baja California), with randomisation of fish to groups. As stated before, this gave an indication of the likely amount of false positives.

**2.3.2 Stock identification.** This analysis attempted to find groups amongst the individuals when no *a priori* grouping has been assumed. This analysis disregarded the location labels for the fish and used only the individuals' marker data. We assessed how well the analysis performed by seeing how many of the individuals have been correctly assigned back to their geographical groups.

The statistical method used was that described in [17] and implemented in the R-package stockR [27]. For this analysis, we specified the number of groups to be four, equal to the number of sampling sites. Once the four groups have been identified, and the probabilities of each individual belonging to each group estimated, the performance of the analysis needs to be evaluated. Here, we assessed performance using the procedure in [17]:

1. Finding the most likely (best) permutation of group labels. This is needed as there is no reason that the ordering of labels will remain consistent between model output and observed data labels (ordering is not invariant within this class of model). As an example: we may have had the individual fish ordered within the groups (Mooloolaba, Solomon Islands, Hawaii, Baja California) but the analysis found a different ordering of (Solomon Islands, Hawaii, Mooloolaba, Baja California). For results to be meaningful, the matching of labels was required.

2. Calculating the number of matches and mis-matches between the best ordering and the real locations.

We note that finding the 'best' ordering may give an optimistic impression of how well the groups were found. The optimistic view may have been heightened with smaller sample sizes—there was a higher chance of a random assignment performing well. The effect of 'chance' good assignment (akin to false positives) was assessed using randomisation of the Western Pacific data where all groups are only nominal.

**2.3.3 Individual assignment.** This analysis attempted to assign an individual to a sampling location based on its genetic data. It was an analysis that is used to identify provenance in a mixed fishery. The analytical method employed here follows that in [18, 19]. This method was chosen as it: 1) allows for the possibility that the individual does not belong to any of the sampled genetic groups, and; 2) is well established in the literature.

For each sampling location, the analysis first calculated a distribution of conditional probabilities (the probability of observing the fish's marker data given the sampling locations' empirical allele frequencies). An individual fish, whose allocation is not known, was inferred to have a genotype that is 'consistent' with a sampling location if its conditional probability was not extreme when compared to the location's empirical distribution [19]. suggested that 'extreme'

be defined as less probable than the 0.01 or even 0.02 percentile. These suggestions are fairly low values, and are used to mitigate the chance of wrongly excluding an individual from the group. Taking such a low value, however, also increases the chance of wrongly suggesting that the individual may come from that (or any other) group. It was not uncommon for the Pacific yellowfin data that multiple groups were consistent with an individual's genetic profile. For this study, we investigated 3 percentile cut-offs: 0.01, 0.05, and 0.1.

Note that the term 'conditional probability', in this analysis, refers to the probability of a genotype assuming that the individual fish comes from a particular location. The assignment method proceeds using a bootstrap method to generate a sampling location's distribution of likely values. See [19] for details.

In our study, this assignment process was repeated for each subsample of the original data. The hold-out fish were assigned to sampling locations based on the data from the subsampled fish. Assignment of an individual fish was considered 'correct' if it was assigned to its actual location, and only that location. The statistic that was summarised within this simulation study was the proportion of correctly assigned fish.

In addition to this assignment—where a fish may not be assigned to any of the sampling locations—we also performed assignment with the assumption that the fish *must* come from one of the sampling locations. To perform this assignment, we made assignments based on the posterior probability of membership statistic defined in [17]. Fish were assigned to the sampling location that had the highest posterior probability.

We assessed the risk of falsely assigning individuals to groups by using data without any real group structure. This was done using randomisation, where fish in the Western Pacific are randomised to groups.

The randomisation and analysis steps we then repeated as before.

## 3 Results

### 3.1 Genetic profile testing

When the number of fish and particularly the number of markers was increased, our ability to delineate locations improves; i.e., there was an increase in the number of rejected hypotheses (lower average p-value; see Fig 3). When a lot of markers was used (our maximum of $m = 4096$), any sample sizes tested proved sufficient to provide excellent power. As the number of markers used decreased, more samples were needed to achieve similar power. For our largest sample size, the minimum number of markers was $m \approx 256$ (Fig 3, left and central panels). Significance appears to be driven largely by Baja California being different from the other sampling locations (Fig 3, centre panel). However, when the differences were smaller, the number of individuals and the number of markers needed to be increased simultaneously (Fig 3, right panel). For those situations (small differences), and with $m \approx 4000$ markers, at least $n \approx 25$ individuals were needed. When testing for differences that were not actually present (randomly assigned groups in the Western Pacific data), the genetic profile test did well. In particular, the average p-value was, as it should have been, around the $\alpha = 0.05$ significance level (Fig 4).

### 3.2 Stock identification

Stock identification for grouping a sample of individual fish using only their genetic information was assessed using the statistic of average proportion of correctly defined stocks (Fig 5). Increased number of markers increased the ability to identify stocks. There was also some counterintuitive evidence suggesting that increasing the number of individuals actually decreases performance. There were two possible reasons for this: 1) there was a noticeable

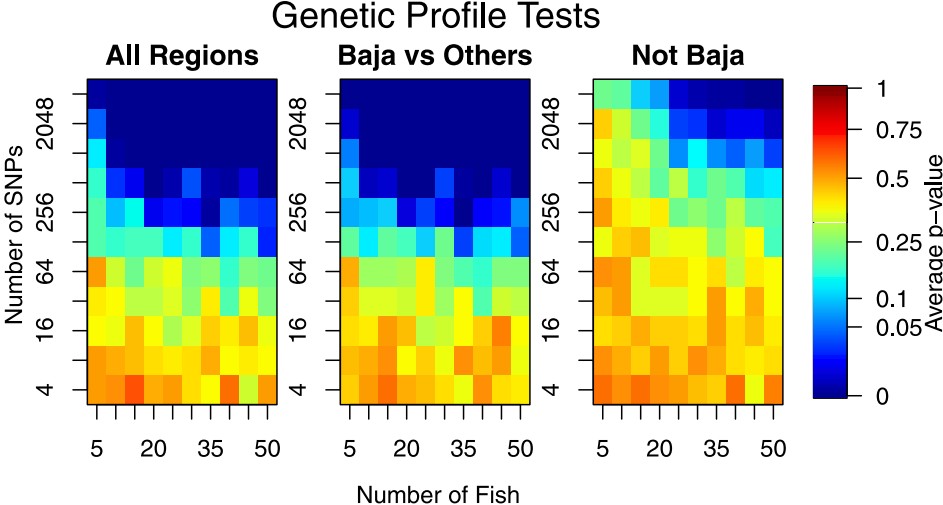

**Fig 3. Results from genetic profile (hypothesis) testing experiment.** Left panel is for the hypothesis test of no differences between all 4 geographical regions. Central panel is for the test of no differences between the Baja California region and the three remaining regions (combined). Right panel is for a test of no differences amongst the three West Pacific regions (Baja California removed). Lower values (cooler colours) reflect lower p-values and hence, on average, a higher chance of rejecting the null hypothesis (i.e., of detecting differences).

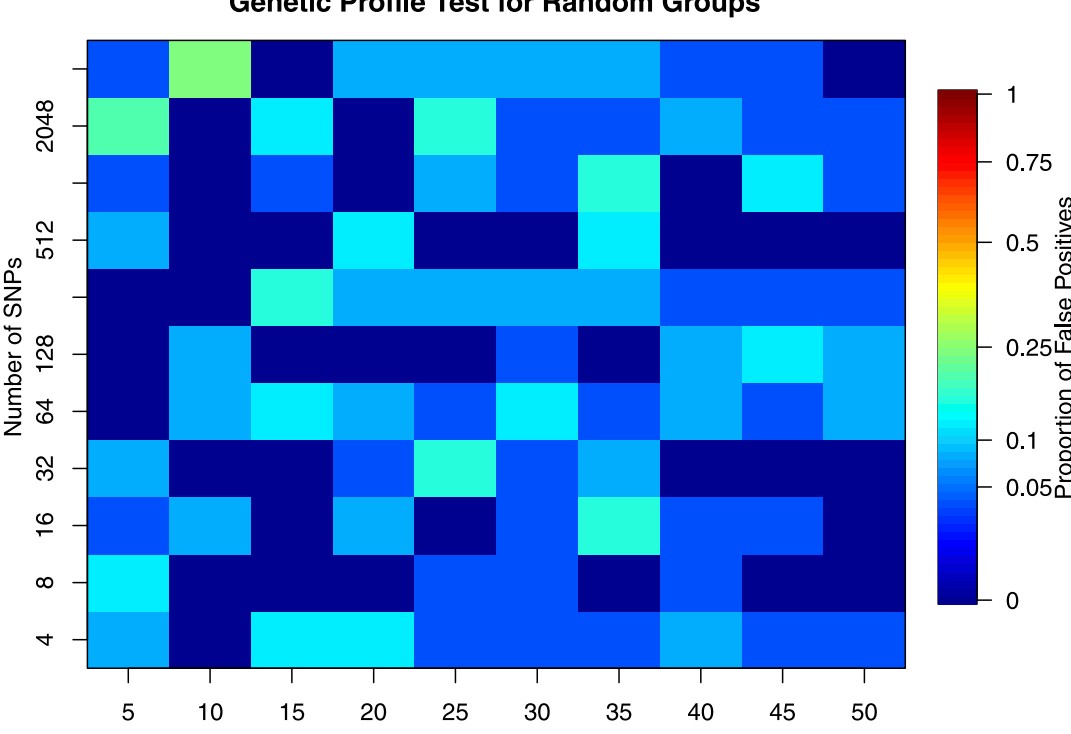

**Fig 4. Results from genetic profile (hypothesis) testing on data with randomised location groups.** The figure contains the proportion of false-positive tests significant at the $\alpha = 0.05$ level.

## Stock Identification

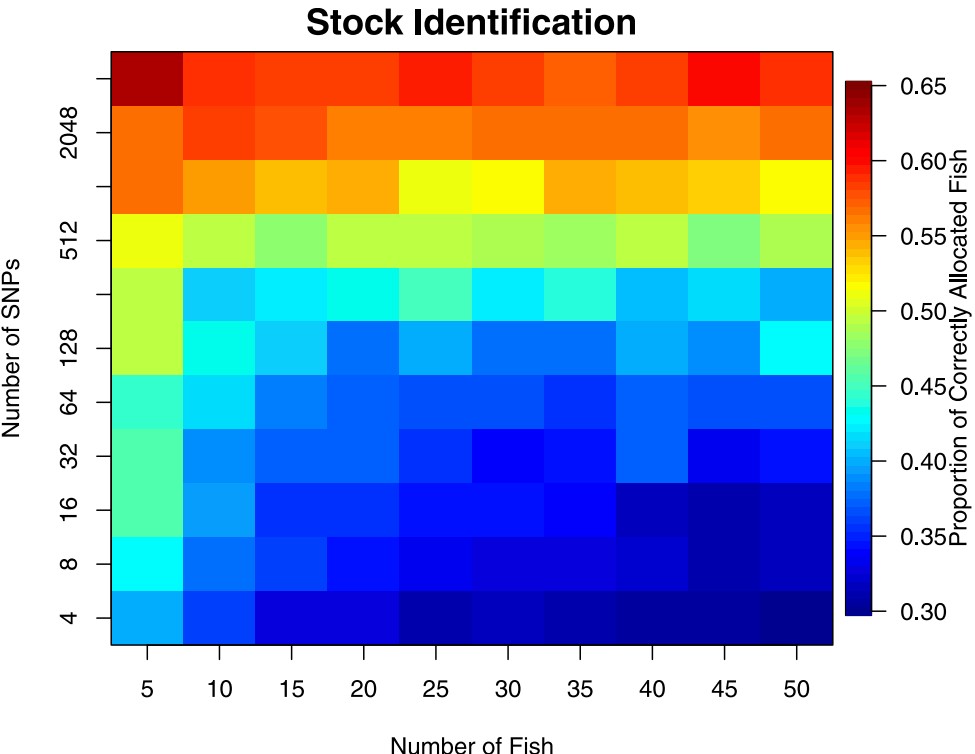

**Fig 5. Results from stock identification analysis from the resampling experiment.** Lower values (cooler colours) represent poorer delineation. This experiment attempted to delineate all four regions.

false-positive rate (Fig 6) and; 2) the 'matching' of the estimated groups to the regions was performed after the analysis (only for performance statistic calculation) and could have biased the metric upwards, and more so for small sample sizes. The false positive rate implied that there is, with small sample sizes, a good chance that any grouping of individuals had an inflated chance to be found.

There appeared to be little difference in using $n = 15$ and $n = 50$ individuals with more than $m = 32$ markers. This implies that $n = 15$ is probably enough. Increasing the number of markers did appear to have substantial effect. Even with $m = 4096$ markers, the benefits of including more appeared to be still increasing.

None of the combinations observed (numbers of individuals and markers) gave excellent discrimination power, with the highest only $\sim 0.75$. This is perhaps unsurprising given that some of the regions were genetically similar (Table 1).

### 3.3 Individual allocation

Results for allocating all the hold-out fish to all sampling locations, and to allocating the hold-out Baja California fish to only the Baja California location are presented in Figs 7 and 8 respectively. In the latter comparison, only the Baja California fish were checked for correct assignment (to Baja California) and fish from other locations were ignored. The broad message is: increasing the number of markers and increasing the number of individuals sampled increased the ability to assign individuals. The most gain was when both markers and individuals were increased. There is little point in trying to allocate individuals to populations when

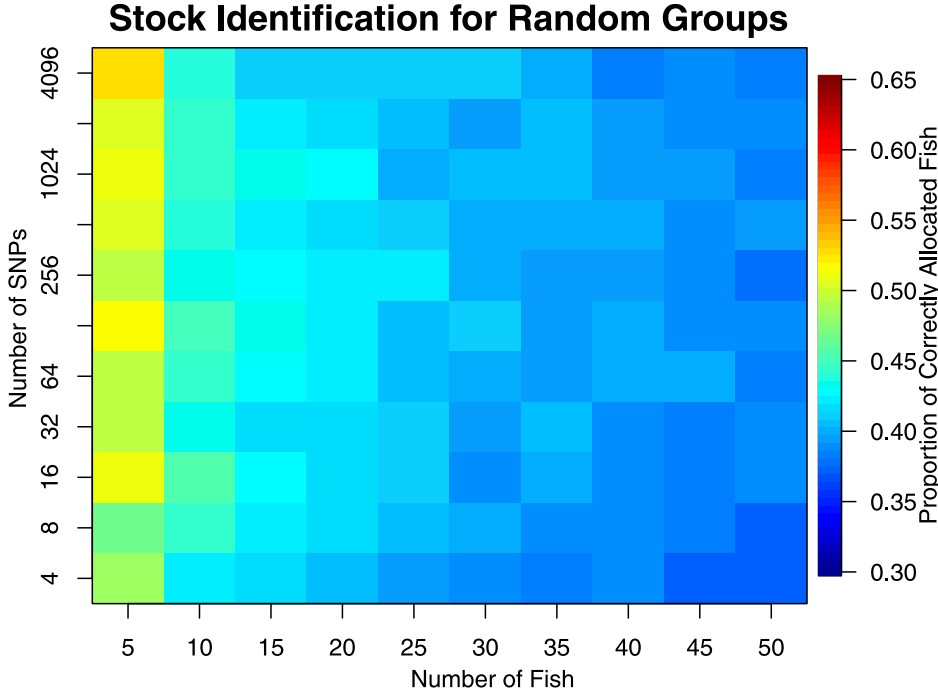

**Fig 6. Results from randomising the location information from the 3 Western Pacific regions for stock identification.** If stock identification is successful, then all values should be 1/K = 1/3. Higher values indicate that groups can be found, even though they are not present. Colour scale chosen to match Fig 5.

there were as few as $n = 5$ or even $n = 10$ individuals, irrespective of the number of markers available.

Assuming that a reasonable number of individuals were sampled, say $n > 25$, then there was more gain by increasing the number of markers than by further increasing the number of individuals sampled from a location (Fig 7, top row). For this increased number of individuals, the performance of assignment increased quite rapidly with increasing markers, after around $m = 256$. At $m = 4096$, the maximum number of markers considered, the assignment performance was still increasing implying that even more markers were likely to have increased performance further. The same inference was made when only allocation to Baja California is considered (Fig 8, top row). However, due to the larger genetic differences, the assignment to only Baja California produced a higher proportion of individuals correctly assigned.

When doing the allocation, we also performed an allocation that assumed that the individual did actually come from one of the observed regions, we just didn't know which one. These results are shown in the bottom panel of Figs 7 and 8. Clearly, allocation was much better when this extra assumption was made; implying that not having an out-group increased performance substantially. However, allocation was still limited when all regions were considered. When only considering the fish from Baja California, the allocation was much improved to when considering all regions.

Randomisation of the Western Pacific data indicated that, irrespective of grouping, assignment can be made with more-or-less the same efficiency as when assigning to the real groups (Figs 7 and 9). The exception was for assignment when assuming that the fish comes from one of these groups (Fig 9 bottom panel). There were two reasons for this: 1) the data available or

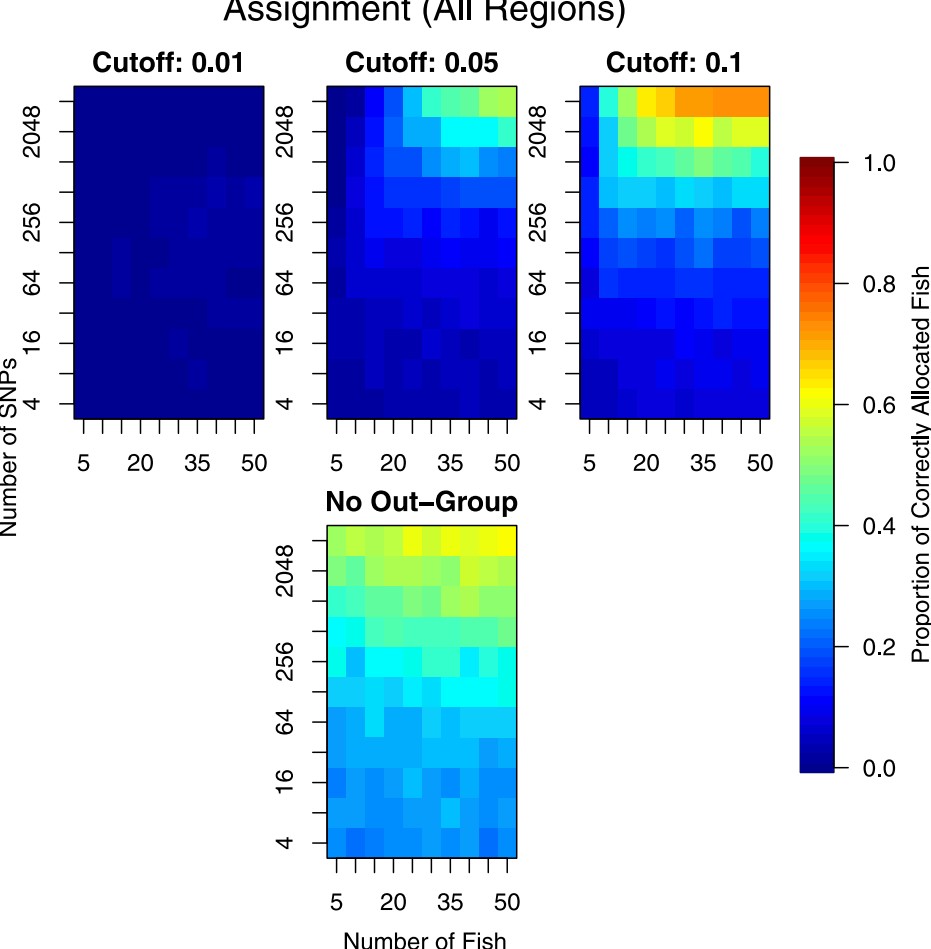

**Fig 7. Results from individual assignment analysis.** Lower values (cooler colours) represent poorer assignment. The top panels are for the same analysis but with different cutoff values, with a smaller cutoff (left) requiring greater dissimilarity to a location before it is considered not to be from that location. The bottom plot is for the situation where the individual was assumed to come from one of the sampled stocks (no chance of it not coming from any of them).

the analytical methods were not sufficient to distinguish the groups, and 2) there was not enough genetic differentiation between the groups to assign with any reliability.

## 4 Summary and discussion

In this investigation, we have examined the effect of increasing the number of individual yellowfin tuna ($n$) and the number of markers ($m$) on the performance of a number of common statistical tasks. This has been done using an extensive data set on yellowfin tuna, which makes the results realistic and without excessive assumptions that are typically made when simulating datasets. Our results showed that small studies were under-powered and were likely to be insufficient for research objectives. However, our results also indicated what type of research questions and the proposed study objectives greatly affected the sample size requirements. Testing genetic profiles between locations required the least information, with only $n \approx 10$ and $m \approx 500$ needed to consistently obtain a significant test (Fig 3). However, if the Baja California region is removed, the remaining populations were less genetically differentiated, requiring a

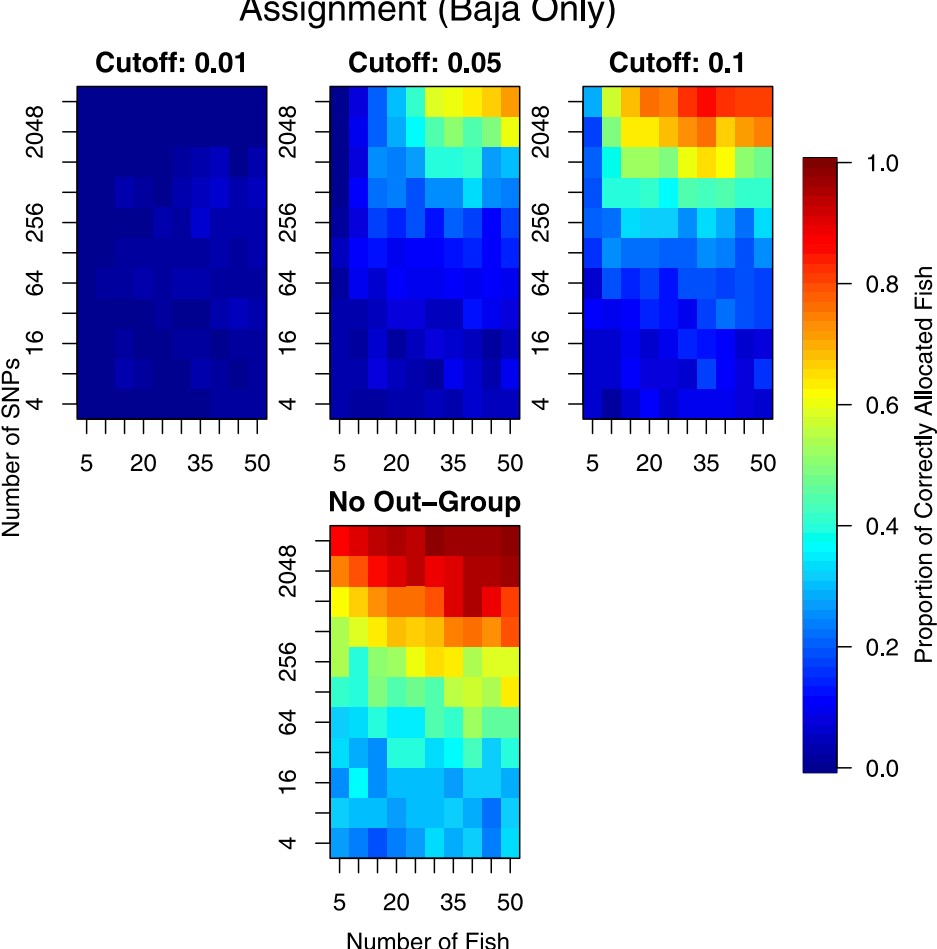

**Fig 8. As per Fig 7, except that assignment is only for fish from Baja California.** These plots reflect how well the Baja California fish are allocated to the Baja California region.

subsequent increase in both sampling effort and numbers of markers investigated ($n \approx 35$ and $m \approx 1000$, Fig 3). Stock identification required more information, with $n \gtrsim 25$ (Figs 5 and 6), and individual allocation required even more information to be reliable (Figs 7–9). In fact, the stock identification and individual allocation were never excellent even for the $n = 50$ individuals and $m = 4096$ markers (Figs 5 and 7–9). This is undoubtedly due to the fact that some of the regions were genetically similar (Table 1).

Random groupings allowed investigation of false inferences. For hypothesis testing, this type of error was approximately as it was specified (the false-positive or type I error rate, Fig 4). For stock delineation, the ability to correctly partition was erroneously increased for small numbers of fish (n $\lesssim$ 30 fish, Fig 6) suggesting more than ~30 fish should be sampled for studies examining stock delineation.

Assignment of fish to groups was not accurate and this was true for all the different numbers of fish and markers investigated (Fig 7). The situation was improved when considering only the most genetically differentiated groups but even this level of differentiation required large sample sizes (Fig 8). Whilst the level of genetic differentiation between the Western Pacific locations was small, it is worrying that a similar amount of assignment success was

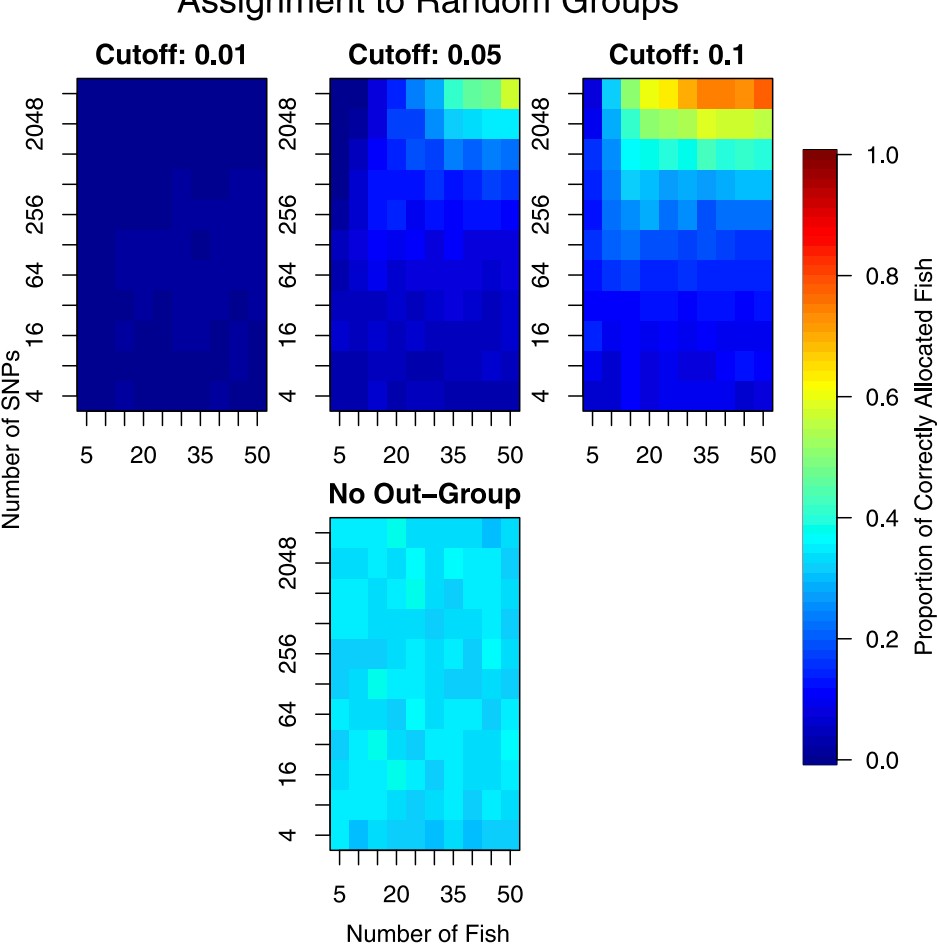

**Fig 9. Results from individual assignment analysis to randomised sampling locations.** Only locations from the Western Pacific (excluding Baja California) are used in this analysis. All other values as per Fig 7. If assignment is random, as it should be, then all values should be 1/K = 1/3.

obtained from randomising fish to locations (Fig 9). This could be explained by a small amount of genetic differentiation, but we note that such differentiation was detectable in group averages (Fig 3, right panel). This result requires further investigation as it may have important implications about the assignment method employed here, or about the data themselves.

When planning sampling for the explicit task of assignment, we note that it is likely to be worthwhile to sample a wide variety of genetic groups. Such sampling is also beneficial for landscape genetic analyses [6]. The reason for choosing a wide variety of genetic groups, for assignment purposes, is that our results showed that the assignment was substantially improved when there was no outgroup (Figs 7–9). In particular, the performance of the assignment to random groups was much better (Fig 9). We acknowledge that there was a potential confounding here between the presence of an outgroup and the analytical method used to perform the assignment. Irrespective of the cause, the outcome remains the same—excluding the outgroup appeared to be more accurate.

For the assignment analysis, there was a natural trade-off between the certainty that we want to exclude individuals from regions and how certain we want to be to include them. This was represented in the different 'cut-off' values. At a cut-off of 0.01 we were asserting that individuals must be less probable than the first percentile of the individuals that do actually belong to that region. This means that an individual-to-be-assigned must be very 'odd' before it was 'rejected' from that population, especially if there was high variance of the genotypes within a population. The consequence of this was that individuals can tend to be 'assigned' to multiple regions. As the cut-off value was increased, the threshold for 'oddness' was decreased, and so we were able to exclude individuals more readily at the risk of false-exclusion from a population. In this resampling study, increasing the cut-off actually increased the assignment capability too.

We note that the results of this study are in line with, but do not duplicate, previous studies. In particular, it is accepted that increasing both the number of individuals and/or the number of markers will increase the success of the analysis (e.g. [6] and references therein). This is achieved by both decreasing bias and decreasing estimate uncertainty [6, Fig 1]. Our results are in line with this as, for example with hypothesis testing, decreased uncertainty corresponds to decreasing p-values (Fig 3).

Ecological genetics is currently undergoing a technological step-change, where the density of information per individual is increased substantially [28]. This doesn't mean that the inferences obtained in this manuscript will be immediately obsolete however. The fact is, RAD genotyping-by-sequencing technology will remain to be utilised into the near future; novel technology delivering higher marker density is still not widely available, especially to most resource-poor projects.

This study assessed the performance of standard statistical methods. It did not compare different methods for the same statistical task. This would have given information about which method gives highest power, or greatest discrimination. While different statistical methods will alter sample size requirements, we hope, and expect, that these changes would not have been substantial. We stress that better, more robust and more powerful statistical methods should be sought, developed and used when available.

Yellowfin tuna is an important species globally, from an economic, food security and a research perspective. These results will aid survey design for new yellowfin tuna studies, whether the study is for profile delineation, stock identification or individual assignment. Researchers wanting to take these results and apply them to other species need to do so with care. Care is needed as differences in evolutionary history and genetic structure (amongst others) will alter the precise data requirements. In such cases, these results should be taken as indicative. When there are no other sources of information, then perhaps surveys should be designed to collect more data than this study suggests.

In this study we purposefully did not simulate surveys which already had pilot study information. If a pilot study was available, then its data could be used in a couple of different ways. Firstly, a pilot study could be used to gauge the level of genetic differentiation present and to see if a small or large number of individuals is required. Secondly, the pilot study could also be used to select a panel/subset of highly informative markers [29] that would then be scored within the main study. This can only be done if the genetic groups were known *a priori*. Since this marker panel is no longer random, the number of markers scored can no longer be inferred from the results of this study.

Finally, we note that there are many possible reasons why adequate sample sizes cannot be obtained, such as when there were no fish available to sample. We note that many of the survey locations are hard to access and the cost of placing a field team into these locations is high. This commits a large proportion of survey budgets to items that only change slightly with

increased numbers of fish and not at all with increased numbers of markers. It is therefore our recommendation that when field work is undertaken, sufficient effort is taken at hard to access locations to collect enough fish to make meaningful scientific inference (as outlined in this study). However, sometimes small sample sizes are truly unavoidable, but does not necessarily mean that a survey should not be performed. Such data may still be valuable, in that it provides some information where there may previously have been none. In the future, it can also provide historical information that would otherwise be completely missing. The key though, it to recognise that inferences may be uncertain, that the uncertainty should be quantified, and incorporate the uncertainty into any conclusions or decisions that stem from the data.

## Supporting information

**S1 Data.**
(RDATA)

**S1 File.**
(ZIP)

## Acknowledgments

We would like to thank: Thierry Gosselin for consistent and diligent help; Paige Eveson for constructive and detailed reading of a preliminary version of this manuscript, and the journal's review team for their efforts.

## Author Contributions

**Conceptualization:** Scott D. Foster, Peter Grewe, Campbell Davies.

**Data curation:** Peter Grewe.

**Formal analysis:** Scott D. Foster.

**Funding acquisition:** Campbell Davies.

**Investigation:** Scott D. Foster, Pierre Feutry.

**Methodology:** Scott D. Foster.

**Supervision:** Pierre Feutry, Peter Grewe.

**Validation:** Scott D. Foster, Pierre Feutry, Peter Grewe, Campbell Davies.

**Writing – original draft:** Scott D. Foster.

**Writing – review & editing:** Scott D. Foster, Pierre Feutry, Peter Grewe, Campbell Davies.

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
