## [Decision Letter · Decision Letter 0]

5 Aug 2021

PONE-D-21-20524

Sample Size Requirements for Genetic Studies on Yellowfin Tuna

PLOS ONE

Dear Dr. Foster,

Thank you for submitting your manuscript to PLOS ONE. After careful consideration, we feel that it has merit but does not fully meet PLOS ONE’s publication criteria as it currently stands. Therefore, we invite you to submit a revised version of the manuscript that addresses the points raised during the review process.

After considering the two contrasting viewpoints from both reviewers, I would like to invite the authors to revise the current manuscript. Although I agree with Reviewer 2 that the data and knowledge brought-forth by the authors deserves publication, Reviewer 1 does provide some valid arguments that I think is worth discussing. Looking forward to the rebuttal/revised manuscript.

We look forward to receiving your revised manuscript.

Kind regards,

Khor Waiho

Academic Editor

PLOS ONE

Journal Requirements:

2. Please amend your list of authors on the manuscript to ensure that each author is linked to an affiliation. Authors’ affiliations should reflect the institution where the work was done (if authors moved subsequently, you can also list the new affiliation stating “current affiliation:….” as necessary)

Additional Editor Comments :

-

Reviewers' comments:

Reviewer's Responses to Questions

**Comments to the Author**

1. Is the manuscript technically sound, and do the data support the conclusions?

Reviewer #1: No

Reviewer #2: Yes

2. Has the statistical analysis been performed appropriately and rigorously? 

Reviewer #1: No

Reviewer #2: Yes

3. Have the authors made all data underlying the findings in their manuscript fully available?

Reviewer #1: No

Reviewer #2: Yes

4. Is the manuscript presented in an intelligible fashion and written in standard English?

Reviewer #1: Yes

Reviewer #2: Yes

5. Review Comments to the Author

Reviewer #1: General comment: In my personal opinion, this paper is not useful and helpful for the community at all for different reasons:

1) It is well established that working with big number of markers (>1K) in population genomics the dataset is not sensitive at all to the number of individuals sampled (Aguirre-Liguori et al 2020). It is clear that the authors are used to work with microsatellite dataset.

2) The field is rapidly moving from restriction site-associated DNA sequencing to low-coverage Whole Genome Resequencing and this is increasing even more our capability to scan the genomes and to obtain more loci and markers. These methods allow to infer complex models of population history even if small sample sizes are available (i.e. 10-15 individuals).

3) For pelagic fish species, it is very complicated to obtain large number of samples. So it Is absolutely useless to point out those numbers for this species. In fact in their paper on the population genomics of yellowfin tuna in the Pacific (Grewe et al., 2015) they have around 15-20 samples per sampling location.

4) It is also complete pointless to have different sample sizes for different task when we are now dealing with more than thousands of markers and we can map them to the genome of many non reference species.

For all these reasons, I would suggest to reject this paper!

Reviewer #2: Comments to the Author:

In this work, authors assessed the numbers of individual yellowfin tuna (Thunnus albacares) and genetic markers required for ocean-basin scale inferences, and they assessed this for three distinct data analysis tasks: testing for differences between genetic profiles; stock delineation, and assignment of individuals to stocks. The results obtained in this work can help designers of molecular ecological surveys for yellowfin tuna to assess whether the information content is adequate for the required inferential task. This work is interesting and very meaningful for researchers. The figures are well presented in the manuscript. Overall, the article deserves publication. However, I also found some shortcomings that require revisions as specified below.

1. There are too many keywords, I think 5-6 keywords are OK. For example, “Single Nucleotide Polymoprhism” is the same with “SNP”.

2. Line 8-9: The first sentence is not very important in the abstract, and it is suggested to delete.

3. In this article, the authors used italics in some sentences, I don’t know the meaning. Such as, “and” in Line 19, “uncertain” in Line 31, “re-sampled” in Line 64, “increasing” in Line 280, and so on.

4. Line 89-90: In order to make it more understandable to readers, it is suggested to mark these four locations in the world map and add this figure to the manuscript.

5. In the Methods section, there is no need to describe the method and the reason so detailedly. For example, Line 99-103, Line 121-125, these parts can be removed to the discussion section. Besides, in the section 2.3 “Statistical Analyses”, is it necessary to depicted it in so detail? The authors can check all the manuscript, and make the Methods and Results sections more succinctly and understandable.

6. In the manuscript, some sentences are not professional and succinctly. For example, “Some researchers may be familiar with this idea as” in Line 149. Please check all the manuscript, and make it more succinctly and professional.

7. In Line 405, delete one “.” after performed.

8. Please check the format of references. For example, “2013, 05” in Line 416, “2012, 09” in Line 446, “;” in Line 454.

6. PLOS authors have the option to publish the peer review history of their article (what does this mean?). If published, this will include your full peer review and any attached files.

Reviewer #1: No

Reviewer #2: No

---

## [Author Response · Author response to Decision Letter 0]

3 Oct 2021

Please see word document "Response2.docx". It has been uploaded with other files and typed as a response to review.

---

## [Editor Report · Decision Letter 1]

13 Oct 2021

Sample Size Requirements for Genetic Studies on Yellowfin Tuna

PONE-D-21-20524R1

Dear Dr. Scott D Foster,

We’re pleased to inform you that your manuscript has been judged scientifically suitable for publication and will be formally accepted for publication once it meets all outstanding technical requirements.

Kind regards,

Khor Waiho

Academic Editor

PLOS ONE

Additional Editor Comments (optional):

The responses provided by the authors are satisfactory and I agree with the authors that it is not feasible to set a fixed (large) sample size for each pelagic fish species or each sampling. There are other influencing factors at play as well. The current version of the manuscript is therefore suitable to be published in PLoS One in its current form. Congratulations.